# Tea Bud and Picking Point Detection Based on Deep Learning

**Junquan Meng** [1], **Yaxiong Wang** [1], **Jiaming Zhang** [1], **Siyuan Tong** [1], **Chongchong Chen** [1], **Chenxi Zhang** [1], **Yilin An** [2] **and Feng Kang** [1,*]

[1] School of Technology, Beijing Forestry University, Beijing 100083, China; yaxiongwang87@bjfu.edu.cn (Y.W.)
[2] School of Foreign Languages, Beijing Forestry University, No. 35 Tsinghua East Road, Beijing 100083, China
[*] Correspondence: kangfeng98@bjfu.edu.cn; Tel.: +86-010-62336137-709

**Abstract:** The tea industry is one of China's most important industries. The picking of famous tea still relies on manual methods, with low efficiency, labor shortages and high labor costs, which restrict the development of the tea industry. These labor-intensive picking methods urgently need to be transformed into intelligent and automated picking. In response to difficulties in identification of tea buds and positioning of picking points, this study took the one bud with one leaf grade of the Fuyun 6 tea species under complex background as the research object, and proposed a method based on deep learning, combining object detection and semantic segmentation networks, to first detect the tea buds, then segment the picking area from the tea bud detection box, and then obtain the picking point from the picking area. An improved YOLOX-tiny model and an improved PSP-net model were used to detect tea buds and their picking areas, respectively; the two models were combined at the inference end, and the centroid of the picking area was taken as the picking point. The YOLOX-tiny model for tea bud detection was modified by replacing its activation function with the *Mish* function and using a content-aware reassembly of feature module to implement the upsampling operation. The detection effects of the YOLOX-tiny model were improved, and the mean average *precision* and *recall* rate of the improved model reached 97.42% and 95.09%, respectively. This study also proposed an improved PSP-net semantic segmentation model for segmenting the picking area inside a detection box. The PSP-net was modified by replacing its backbone network with the lightweight network MobileNetV2 and by replacing conventional convolution in its feature fusion part with Omni-Dimensional Dynamic Convolution. The model's lightweight characteristics were significantly improved and its segmentation accuracy for the picking area was also improved. The mean intersection over union and mean pixel accuracy of the improved PSP-net model are 88.83% and 92.96%, respectively, while its computation and parameter amounts are reduced by 95.71% and 96.10%, respectively, compared to the original PSP-net. The method proposed in this study achieves a mean intersection over union and mean pixel accuracy of 83.27% and 86.51% for the overall picking area segmentation, respectively, and the detecting rate of picking point identification reaches 95.6%. Moreover, its detection speed satisfies the requirements of real-time detection, providing a theoretical basis for the automated picking of famous tea.

**Keywords:** target detection; tea bud; picking point; semantic segmentation

## 1. Introduction

Tea is one of the most important beverages in the world and a profound tea culture has formed in all continents, featuring a wide audience, high demand and high economic value [1]. Currently, mechanical picking methods have been applied in the field of tea picking, but the tea leaves picked by these method are not complete, and there are a large number of old leaves and broken branches mixed in it, so mechanical methods are only used for bulk tea picking with low economic value [2]. Famous tea has high nutritional and market value and rare production; famous tea strictly requires quality picking, so it still relies on manual picking with low efficiency, high cost and labor shortages, which have

become important problems restricting the development of the famous tea industry [3]. In order to promote the development of this industry, there is an urgent need to improve the working mode of famous tea picking and transform it into an intelligent way, i.e., replace manual picking with intelligent machine picking. The key breakthrough of intelligent picking is the automatic recognition of tea buds and their picking points in complex scenes. Compared with other target detection tasks [4–6], the color of tea buds is close to that of the background, the sizes of targets are small and densely distributed, and the similarity between tea bud individuals is not high enough, and therefore, these characteristics increase the difficulty of tea bud detection.

Currently, research in this field is mainly divided into recognition of tea buds and detection of tea bud picking points. The methods adopted are mainly divided into traditional image segmentation and deep learning. Studies on tea bud recognition through traditional segmentation methods include the following: Zhao et al. [7] segmented tea buds by setting a threshold to combine three channels under HSV color space and the method achieved good results. Shao et al. [8] used the S factor under HSI color space to carry out gray level analysis of images, and then the improved K-means algorithm was adopted to extract the tea buds. Traditional segmentation methods used for picking point extraction include the following: Long et al. [9] extracted the ultra-green features of images; the Otsu method was used for threshold segmentation to obtain the segmented tea bud regions and the edge detection and skeleton extraction methods were combined to locate the picking points. Lei et al. [10] extracted the G-B feature of tea buds and used the Otsu method for secondary segmentation to extract tea bud skeleton. The Shi–Tomasi algorithm was used to detect the skeleton corners and to mark the picking points, and the recognition rate was 85.12%. Traditional segmentation methods use the difference in color between the tea buds and the background to extract targets, and have relatively simple algorithms and small amounts of computation, whereas they are poor in robustness and are difficult to detect in real time.

With the continuous development of deep learning, convolutional neural networks are gradually being applied to the field of tea bud detection; the essential feature of deep learning is its great self-learning capability and strong perception of similar features. Some studies have used object detection algorithms to detect tea buds. Xu et al. [11] used the YOLOv3 algorithm to detect tea buds and the DenseNet201 algorithm to further classify them, and the detection accuracy of the method was 95.71%. Cao et al. [12] replaced the backbone network of YOLOv5 with GhostNet, and BiFPN structure was adopted in the feature fusion part, and the accuracy of tea bud detection was 76.31%. Some studies have used object detection or image segmentation algorithms to detect tea bud picking points. Chen et al. [13] used a Faster R-CNN to detect the OTTL regions, and then used a full convolutional network (FCN) to identify picking points within the OTTL regions. The *precision* of the Faster R-CNN model was 79% and the *recall* rate was 90%; the FCN achieved an average accuracy of 84.91% and an *mIoU* of 70.72%. Yan et al. [14] proposed an improved Mask R-CNN model to detect the picking points; the *mAP* value of tea bud recognition was 44.9%, the f2 value was 31.3%, and the positioning accuracy of picking points was 94.9%. Some studies have combined deep learning and traditional image segmentation methods, fusing the advantages of two methods, that is, the lightweight computation of conventional image segmentation and strong learning ability of deep learning. Yang et al. [15] used an improved YOLOv3 model to detect tea bud regions, extracted the foreground and the skeleton of tea buds through morphological processing, and the lowest point of the minimum rectangle of the skeleton was selected as the picking point. Zhou et al. [16] replaced the backbone network of YOLOv3 with DenseNet, and the object surface was obtained by using the depth threshold method after detecting tea buds, the skeleton of tea buds was obtained by connecting the boundary feature points, the optimal shear points were determined according to the morphology of tea buds. Yan et al. [17] used an improved DeeplabV3+ semantic segmentation algorithm to segment the foreground of tea buds, and obtained the maximum connected domain of the target after binarization. Finally, the

Shi–Tomasi algorithm was used to calculate the corner of the maximum connected domain, and the corner with the lowest ordinate was marked as the picking point.

Aiming at the problems that the background of tea buds in natural state is complex and it is difficult to identify the bud and select the picking point, this study took the one bud with one leaf grade of the Fuyun 6 tea species under complex background as the research object, which was close in color to the background, including different growing stages and postures. We proposed a tea bud and picking point identification method based on deep learning, combining object detection and semantic segmentation algorithms. This study innovatively chose the tea bud which had more apparent characteristics as the region of interest, and proposed to detect the tea bud first, then to segment the picking area within the tea bud detection box, and finally to obtain the picking point within the picking area. A lightweight YOLOX-tiny model was selected for tea bud target detection and the PSP-net semantic segmentation algorithm was used to segment the picking areas; the two models were combined together at the inference end after they were modified. The improved YOLOX-tiny model was used to predict the original images, to obtain the tea bud detection boxes, and the boxes were input into the improved PSP-net for postprocessing, to obtain the picking area. Finally, the centroid point of the segmented picking area was selected as the picking point.

## 2. Materials and Methods

### 2.1. Experimental Equipment and Dataset Processing

The experimental hardware utilized in this study consisted of an AMD Ryzen5 3600X CPU and a NVIDIA GeForce RTX 3060 Ti graphics processing unit. The operating system utilized was Windows 10 and the experiments were conducted within the Anaconda environment. Pytorch1.10.0 was utilized as the deep learning framework. A total of 300 epochs were implemented for training, utilizing pretrained weights. The hyperparameter settings for the training process are presented in Table 1.

**Table 1.** Hyperparameter settings for the training of models.

| Freezing Epoch | Freezing Batch Size | Unfreezing Epoch | Unfreezing Batch Size | Optimizer | Initial Learning Rate | Momentum | Weight Decay | Learning Rate Decay Type |
|---|---|---|---|---|---|---|---|---|
| 50 | 4 | 300 | 2 | SGD | $1 \times 10^{-2}$ | 0.937 | $5 \times 10^{-4}$ | Cos |

The experiment site in this study was the Wangu Tea Garden in Mingliang Town, Shanglin County, Nanning City, Guangxi Zhuang Autonomous Region (23°19′38″ N, 108°39′24″ E). The shooting equipment used was a Huawei Mate 30 smartphone with a rear camera (a 40-megapixel main camera and an 8-megapixel telephoto camera). The Fuyun 6 tea species was selected as the research object. The best time for tea bud picking is early April [18], so dataset shooting was conducted from 4–7 April 2022. Images were shot at a horizontal angle under strong light (12:00–14:00) and weak light (17:00–19:00). A total of 1063 original images were captured, including 533 images taken under strong light and 530 images under weak light; all images were saved in JPG format. This study produced three datasets, including (a) a tea bud detection dataset, (b) a semantic segmentation dataset of the picking area and (c) a directly detecting picking area dataset. The detailed information of the three datasets is listed in Table 2.

**Table 2.** Detailed information of three datasets.

| | Labeling Tool | Labeling Method | Augment Method | Number of Original Dataset | Number of Final Dataset | Training Set | Validation Set | Test Set |
|---|---|---|---|---|---|---|---|---|
| (a) Tea bud dataset | Labelimg | Labeled on the original images | Horizontal mirroring | 1063 | 2126 | 1360 | 340 | 426 |
| (b) Picking area dataset | Labelme | Labeled within the detection boxes | | 1153 | 2306 | 1475 | 369 | 462 |
| (c) Directly detecting picking area dataset | Labelme | Labeled on the original images | | 1063 | 2126 | 1360 | 340 | 426 |

### 2.1.1. Labeling Method for Tea Bud Detection Dataset

One bud with one leaf is usually recognized as first-class tea with high economic value [19]. The standard of one bud with one leaf, namely the tender bud and the first leaf next to it, was labeled in this study. There are two forms of one bud with one leaf: spreading and developed [20]. The two forms are not distinguished during actual picking, and both are targets to harvest, so this study identified both spreading and developed forms as objects to recognize. As shown in Figure 1, the tea buds were divided into four classes for labeling.

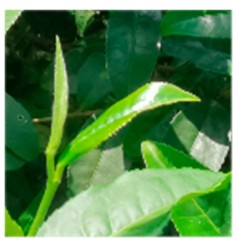
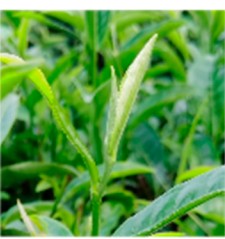
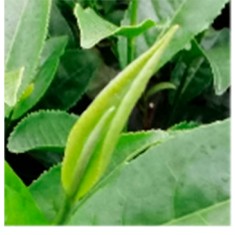
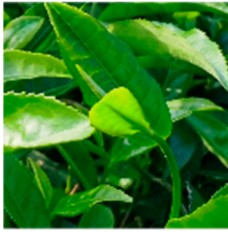

(**a**) Class "1"          (**b**) Class "2"          (**c**) Class "3"          (**d**) Class "c"

**Figure 1.** Labeling method for tea bud detection dataset: (**a**) Class "1", stretched, bud and leaf are clearly separated and the first leaf is completely stretched; (**b**) Class "2", maturely stretched, bud and the first leaf are not completely separated and the leaf is curled up; (**c**) Class "3", stretched, with buds contained within the first leaf; (**d**) Class "c", buds in lateral posture.

### 2.1.2. Labeling Method for Picking Area Dataset

After training and prediction using the improved YOLOX-tiny model on the original images, 1153 tea bud detection boxes were selected and cropped to create a dataset for the picking area. Annotations were made within the detection boxes output from the improved YOLOX-tiny model, as shown in Figure 2a. Referring to the position of the bud-pulling point selected during manual tea picking, the tea stem portion within the tea bud detection box was regarded as the picking area and annotated accordingly.

### 2.1.3. Labeling Method for the Directly Detecting Picking Area Dataset

In order to explore the better method for detecting picking points, in this study, we compared the effects of our method and the method of directly detecting the picking area on the original images. Directly detecting the picking area means that the tea stem areas on the original images are directly segmented by the segmentation model, without any other preprocessing. A segmentation dataset of directly detecting the picking area was produced and labeled on the original images, as shown in Figure 2b. According to the actual picking, the tea stem area within about 3 mm below the end of the tea bud was regarded as the picking area.

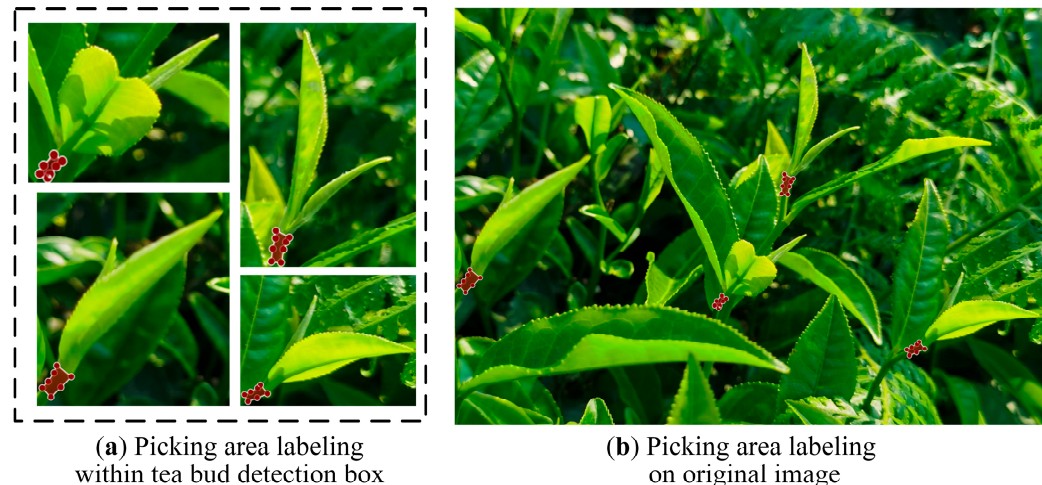

**(a)** Picking area labeling
within tea bud detection box

**(b)** Picking area labeling
on original image

**Figure 2.** Labeling method for the picking area dataset and the directly detecting picking area dataset: (**a**) The picking area dataset was labeled on the detection boxes output from the improved YOLOX-tiny; (**b**) the directly detecting picking area dataset was labeled on the original images.

### 2.2. Picking Point Detection Method Combining the Improved YOLOX-Tiny and the Improved PSP-Net

A target detection model identifies and locates targets through detection boxes, and a semantic segmentation model identifies and locates targets through the segmented pixel areas. A target detection model can provide the accurate location and size information of small-sized targets, while the accuracy of a semantic segmentation model for small-sized targets is relatively poor. Considering the characteristics that tea buds are small in size and densely distributed, and their picking areas are not regular rectangles, whereas compared to picking areas, tea buds are more differentiable to the background, we proposed an idea to detect the buds first, then secondly, to obtain the picking area based on the tea bud detection boxes, and finally, to obtain the picking point based on the picking area. We decided to use a target detection model to detect tea buds, and a semantic segmentation model to segment the picking area, the picking point is within the picking area. In this study, we combined a target detection model (the improved YOLOX-tiny) and a semantic segmentation model (the improved PSP-net) for tea bud picking point detection. The workflow is shown in Figure 3; the improved YOLOX-tiny was used to train the tea bud detection dataset, and the improved PSP-net was used to train the picking area dataset. The two models were combined at the inference end, the cropped tea bud detection boxes were output from the improved YOLOX-tiny, and then they were input into the improved PSP-net to segment the tea stem area within the detection boxes as the picking area. Finally, the centroid pixel point of the picking area was regarded as the picking point.

### 2.3. Tea Bud Detection Method Based on YOLOX-Tiny

#### 2.3.1. The Improved YOLOX-Tiny Model

YOLOX [21] was proposed in 2021, and like its predecessors, it is divided into backbone, neck, and head parts, as shown in Figure 4. YOLOX uses CSPDarknet as its backbone network, continues to use Feature Pyramid Network (FPN) structure in the neck part, and uses a decoupled head with stronger expression ability in the head part, combined with Mosaic data enhancement, anchor free and SimOTA methods, to achieve stronger detection ability and faster convergence speed, with high accuracy on public datasets and fast detection speed. To facilitate the deployment of this algorithm on mobile devices for real-time tea picking, a lightweight model should be selected. In this study, the YOLOX-tiny model with lower computation and parameter amounts and relatively high detection accuracy was selected for tea bud detection. To improve the detection ability of the model and to avoid introducing a lot of extra computational effort, this study did not add additional modules but replaced the existing modules of the original model. We introduced the *Mish*

activation function to replace the Silu activation function, and a content-aware reassembly of feature (CARAFE) module was introduced to perform the upsampling operation.

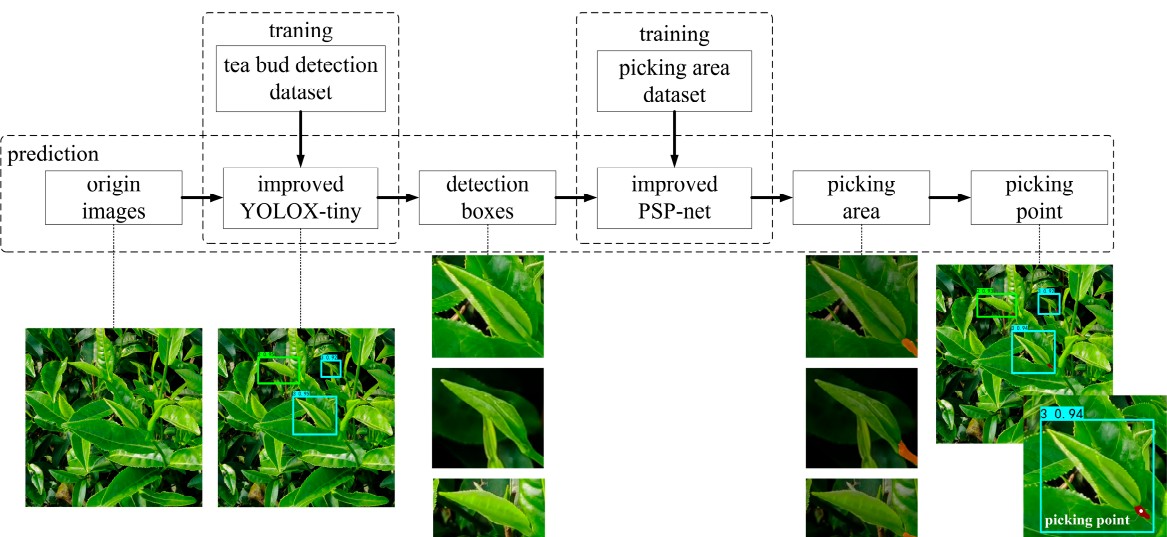

**Figure 3.** Schematic diagram of the detection method for tea bud picking point.

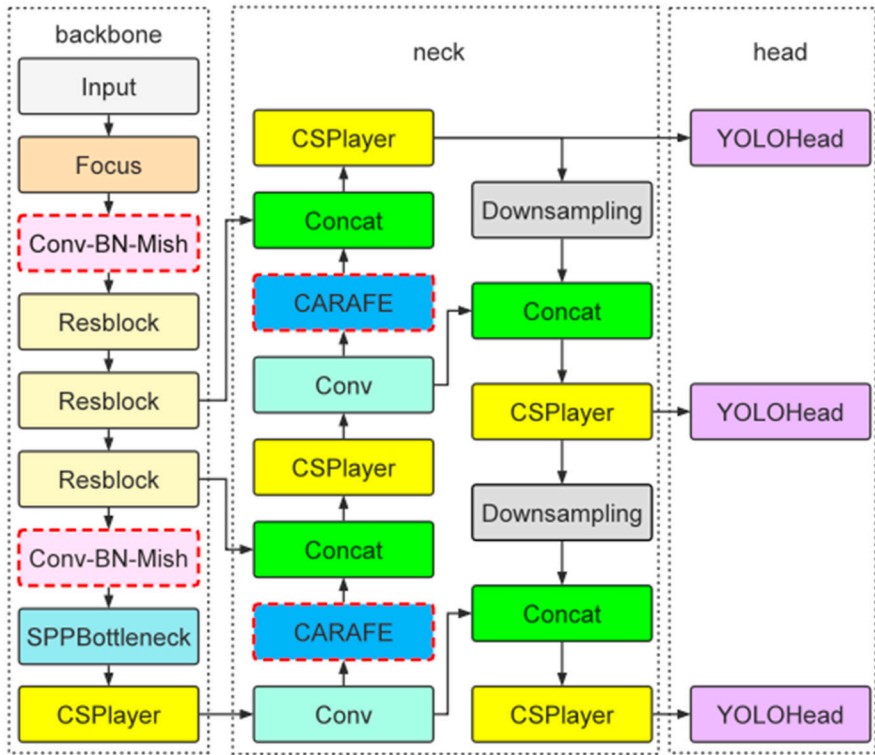

**Figure 4.** Network structure of YOLOX-tiny and its improvement. The black boxes are modules of the original YOLOX-tiny, the red dotted boxes are the modified modules, the modules within parenthesis are utilized.

### 2.3.2. *Mish* Activate Function

The *Mish* activation function [22] is a nonlinear function with an upper bound and no lower bound, and it is not monotonically increasing or decreasing in its domain of definition.

When the input is a small negative value, *Mish* outputs a similar small negative result; a small negative input can also activate its neurons for fitting. Except when the input tends to negative infinity, the gradient tends to zero, the *Mish* function for other values of input can maintain the stability of the gradient flow, which efficiently prevents the phenomenon of gradient disappearance. The curve of the *Mish* function is smooth to achieve better generalization and propagation ability. The *Mish* function is defined as follows:

$$Mish(x) = x \times \tanh(\ln(1 + e^x)), \tag{1}$$

$$\tanh(t) = \frac{e^t - e^{-t}}{e^t + e^{-t}}, \tag{2}$$

The Silu activation function is used in the original YOLOX-tiny model. In this study, the *Mish* activation function was introduced to compare the detection effects with the predefined Silu function and to select the function with better effects as the activation function.

### 2.3.3. The CARAFE Module

Since tea shoots are relatively similar to each other and the targets are small in size, to meet the requirement of more detailed features, this study used a content-aware reassembly of feature (CARAFE) [23] module for the upsampling operation, which helped to retain more detailed features of tea buds and obtain higher quality images even after the size was enlarged.

YOLOX-tiny uses a nearest-neighbor interpolation-based upsampling module. In this study, we introduced the CARAFE module to implement the upsampling operation. Unlike the interpolation-based upsampling operation, the CARAFE method makes use of the semantic information of the feature map. The CARAFE consists of two modules, i.e., upsampling kernel prediction and feature reorganization. The upsampling kernel prediction module of CARAFE consists of three main components, namely channel compression, content encoding and kernel normalization. The CARAFE utilizes its larger perceptual domain to make full use of the surrounding feature information, and its upsampling kernels are related to the semantic information of input, focusing more on local point information than interpolation, and the upsampling is based on the content of the feature map rather than just its position, providing richer and higher leveled semantic information. The CARAFE has better performance than traditional upsampling operations and its lightweight characteristic avoids introducing a large amount of computational cost.

### 2.4. The Improved PSP-Net Model

The structure of the original and improved PSP-net [24] is shown in Figure 5. For the input image, the feature map is extracted by the backbone network ResNet, and the output feature map is one-eighth of the size of the original image. The pyramid pooling module in Figure 5d is used to obtain the contextual information of the feature map, where the pyramid pooling module is divided into four layers as shown in the figure, and finally, the four layers can be fused into global features. The obtained global features are stitched together with the original feature map through a residual structure, and a convolution layer is used to generate the final prediction map in Figure 5e. The most important characteristic of PSP-net is the usage of the pyramid pooling module, where the obtained feature maps are divided into different sizes and numbers of subregions and the average pooling operation is performed within each subregion, which effectively produces high-quality results in scenario analysis tasks, thus improving the ability of obtaining global information. PSP-net provides effective global contextual information for pixel-level scene parsing. In order to make it more suitable for the segmentation task in this study, the model should be lightweight enough with segmentation accuracy guaranteed. The PSP-net model was modified by replacing the backbone network with MobileNetV2 and by incorporating Omni-Dimensional Dynamic Convolution (ODConv). The improved PSP-net model was used to segment the tea stem part within the detection boxes output from YOLOX-tiny.

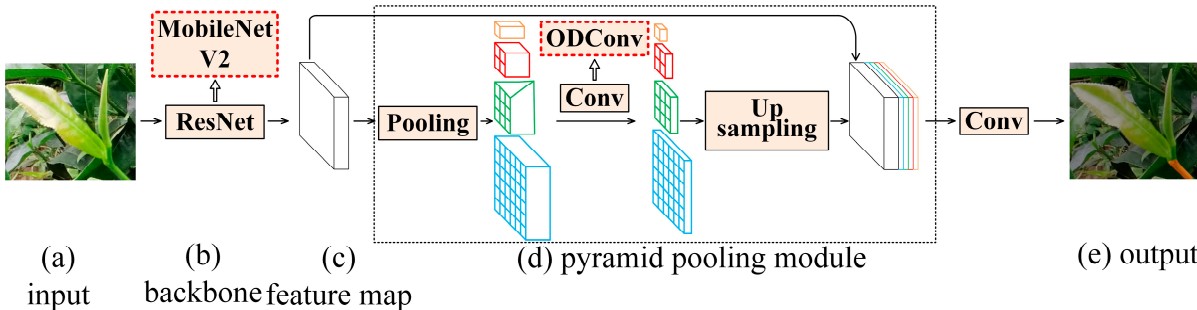

Figure 5. The network structure of PSP-net and the improved model: (**a**) The input image;
(**b**) backbone network; (**c**) feature map output from backbone; (**d**) the pyramid pooling module;
(**e**) the output image.

### 2.4.1. MobileNetV2 Backbone Network

The segmentation task in this study is relatively simple, and does not require a large
volume of the model. In order to meet the requirements of real-time detection and to
avoid wasting arithmetic power, in response to the large volume and slow computing
speed of the PSP-net model, in this study, the original ResNet backbone network was
replaced by the lightweight network MobileNetV2 [25], which could significantly reduce
the number of parameters and improve the detection speed of the model. The structure of
the MobileNetV2 backbone network is shown in Table 3.

**Table 3.** The network structure of the MobileNetV2 backbone.

| Input Size | Operation | Expansion Coefficient | Input Channels | Output Channels | Convolution Times | Stride |
|---|---|---|---|---|---|---|
| 512 × 512 | Conv2d | - | 3 | 32 | 1 | 2 |
| 256 × 256 | bottleneck | 1 | 32 | 16 | 1 | 1 |
| 256 × 256 | bottleneck | 6 | 16 | 24 | 2 | 2 |
| 128 × 128 | bottleneck | 6 | 24 | 32 | 3 | 2 |
| 64 × 64 | bottleneck | 6 | 32 | 64 | 4 | 2 |
| 32 × 32 | bottleneck | 6 | 64 | 96 | 3 | 1 |
| 32 × 32 | bottleneck | 6 | 96 | 160 | 3 | 1 |
| 32 × 32 | bottleneck | 6 | 160 | 320 | 1 | 1 |

MobileNet [26] is a lightweight deep neural network proposed by Google in 2017,
featuring small size, high accuracy and fast computing speed, whose core idea is the
usage of depthwise separable convolution. MobileNetV2 is the upgraded version, and
the most highlighted improvements to MobileNet are the usage of inverted residuals
and linear bottlenecks. Compared to MobileNet, MobileNetV2 uses channel compression
followed by expansion, inverted residuals first increase the number of channels, allowing
the depthwise separable convolution to extract more features before channel compression.
This modification improves the phenomenon that depthwise separable convolution does
not extract enough features after compressing the number of channels, and optimizes the
feature extraction capability of MobileNet. The ReLU activation layer is replaced with a
linear layer in the final layer of the bottleneck structure, to compensate for the tendency of
the ReLU function to lose negative-value features.

### 2.4.2. Omni-Dimensional Dynamic Convolution

In order to enhance the feature extraction capability of the model, to enrich the expression information of tea bud features, to compensate for the loss of accuracy caused by the
lightweight backbone networks and to avoid introducing a large amount of computational
cost, omni-dimensional dynamic convolution (ODConv) [27] was introduced in this study
to replace the 3 × 3 convolution in the PSP module.

The main idea of dynamic convolution is to use different convolution kernels for different inputs, and then weigh these kernels with attention. ODConv summarizes the features of CondConv [28] and DynamicConv [29], building on them to make improvements. On top of weighting the number of convolution kernels, ODConv introduces three additional dimensions, namely the number of input channels, the number of output channels and the perceptual domain of the convolution kernel. ODConv utilizes a novel multidimensional attention mechanism and parallelism strategy to weigh all dimensions of the convolution kernel. These four types of attention are complementary, and by progressively multiplying the convolution along the position, channel, filter and kernel dimensions with different attentions will allow the convolution operation to differ across dimensions for the input, better capturing rich contextual information. Thus, ODConv can improve the feature extraction capability of convolution, and ODConv with fewer convolution kernels can achieve better performance than CondConv and DyConv. ODConv can be conveniently inserted into many CNN architectures, and experimental results show that it can improve the performance of both large- and lightweight models.

## 3. Results

As shown in Figure 6, in this section, the indicators used to evaluate the performance of each model are first introduced in Section 3.1. In the second part of this section, Section 3.2.1 presents the comparative experimental results of different target detection models, explaining that YOLOX-tiny is the most suitable tea bud detector for our method and Section 3.2.2 presents the results of each step of modification of the improved YOLOX-tiny. Similarly, the third part of this section firstly presents the comparative experimental results of different semantic segmentation on picking area segmentation within the tea bud detection box in Section 3.3.1, explaining that the PSP-net model is the most suitable. Section 3.3.2 presents the results of each step of modification of the improved PSP-net model. Section 3.4 presents the results of our method, combining the improved YOLOX-tiny and the improved PSP-net models, on picking area segmentation on the original images, and compares them to the method of directly segmenting the picking area on the original images. Different to the above chapters, Section 3.5 presents comparisons through actual images and the analyses of the actual detection effects of YOLOX-tiny and its improved model according to the detected images, the actual segmentation effects of PSP-net and its improved model according to the segmented images, and the actual picking area segmentation effects on the original images of our method and the directly segmenting method according to the actual segmented images. In addition, the actual selection effects of the picking points after the picking area are determined.

### 3.1. Evaluation Indicators

3.1.1. Evaluation Indicators for Target Detection Models

This study evaluates the detection accuracy of the target detection models using mean average *precision* (*mAP*) (%) and mean *recall* (*mR*) (%), the detection speed using frame per second (*FPS*) (frames), and the volume of the model using computation amount GFLOPS (G) and parameter quantity PARAMS (M). *mAP* is calculated from average *precision* (AP) and the number of classes, and AP is calculated from *precision* (P) and *recall* (R). *mR* is calculated from *recall* and the number of classes. *Precision* (%) means the percentage of targets predicted by the model that are actually targets, and it evaluates how accurate the model is in recognizing targets; *precision* (%) is calculated using the following equation:

$$Precision = \frac{TP}{TP + FP},\qquad(3)$$

*Recall* (%) represents the percentage of targets identified by the model to the number of real targets, and it evaluates how many of all targets the model has identified. *Recall* (%) is calculated using the following equation:

$$Recall = \frac{TP}{TP + FN},$$ (4)

where *TP* (true positive) indicates the correctly identified true target, i.e., a target is identified as a target; *FP* (false positive) indicates the incorrectly identified target, i.e., a background is identified as a target; *TN* (true negative) indicates the correctly identified background, i.e., a background is identified as a background; *FN* (false negative) indicates the incorrectly identified background, i.e., a target is identified as a background. Here, *TP*, *FP*, *TN* and *FN* evaluate each of the detection boxes [30].

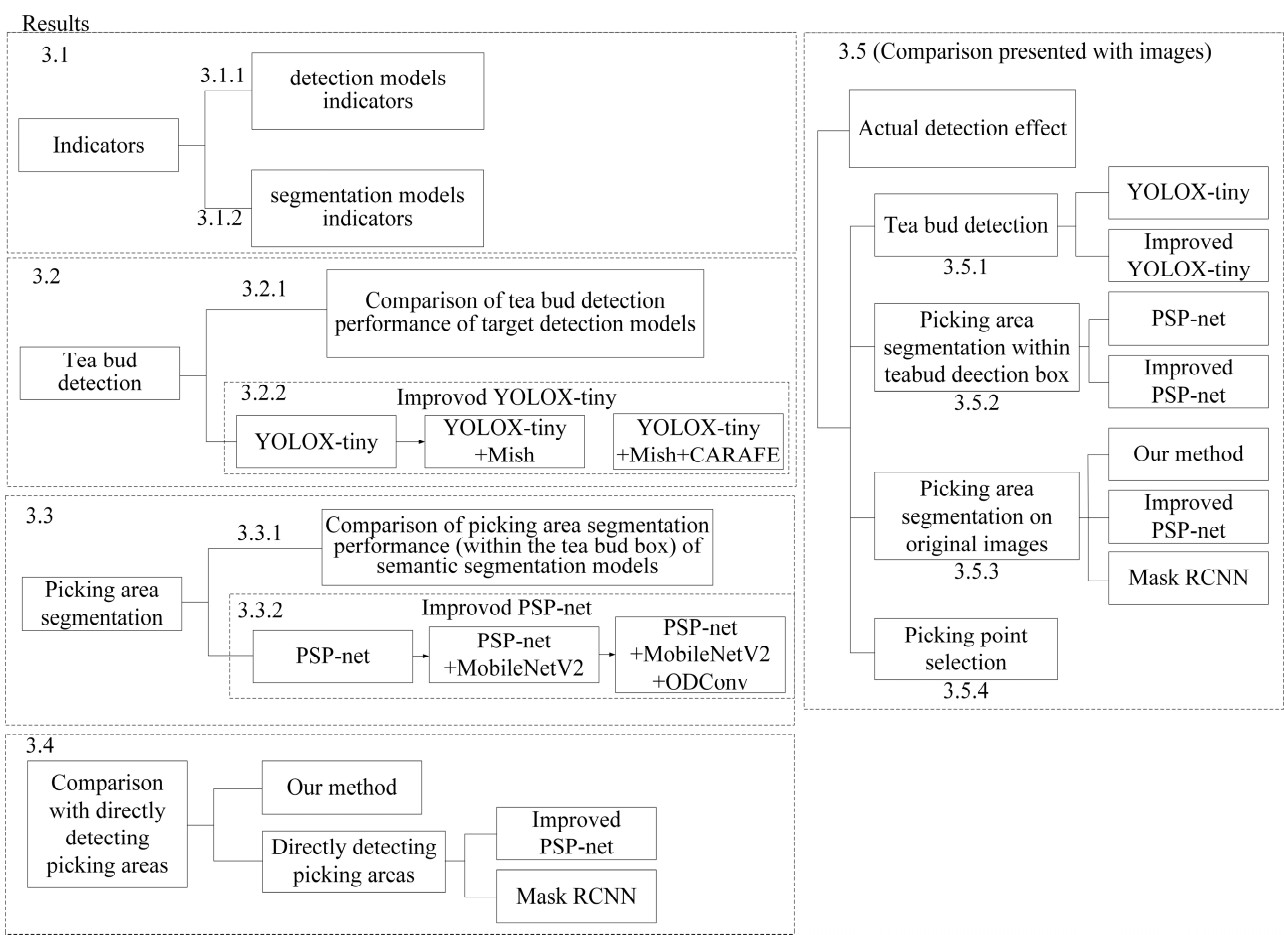

**Figure 6.** The logic structure of the Results section.

Since *precision* (%) and *recall* (%) have interactive effects and different *precision* (%) and *recall* (%) values can be calculated from different confidence thresholds, it is difficult to evaluate the two indicators comprehensively, so the area enclosed by the P-R curve formed by the two indicators is used to comprehensively evaluate the detection accuracy of the model [31], expressed as *AP* (%):

$$AP = \int_0^1 P(R)dR,$$ (5)

where *AP* (%) indicates the average *precision* of the model for a single class of targets. The higher this value is, the more accurate the model is in detecting targets in that class.

*mAP* (mean average *precision*) (%) represents the mean value of *AP* (%) for all classes (assuming k classes), and the higher its value is, the more accurate the model is in detecting targets in all classes. *mAP* (%) is calculated by the following equation:

$$mAP = \frac{\sum_{1}^{k} AP(i)}{k}, \tag{6}$$

*Recall* (%) indicates the model's detection completion rate for a single class of targets, and the higher its value is, the more accurate the model is in detecting targets in that class. The *mR* (mean *recall*) (%) represents the mean value of *recall* (%) for all classes (assuming *k* classes):

$$mR = \frac{\sum_{1}^{k} R(i)}{k}, \tag{7}$$

*FPS* (frames) indicates the number of frames processed per second and is calculated as follows:

$$FPS = \frac{1}{T}, \tag{8}$$

where *T* (s) is the time taken by the model to process an image.

The computation amount (GFLOPS) (G) and parameter quantity (PARAMS) (M) represent the volume of the model, where GFLOPS (G) is influenced by the number of layers, depth, number of channels, input image size, total number of target classes, etc., and PARAMS (M) represents the size of the memory space occupied by the model. The higher the values of these two indicators are, the larger the model is and the greater the computing power requirement.

3.1.2. Evaluation Indicators for Segmentation Models

In this study, the segmentation accuracy of the segmentation models is evaluated using the mean pixel accuracy (*mPA*) (%) and the mean intersection over union (*mIoU*) (%). *mPA* is calculated from pixel accuracy (PA) (%) and the number of classes. Pixel accuracy (%) is based on the pixel level and represents the proportion of correctly predicted pixels to the total pixels. *mPA* (%) is the mean value of *PA* (%) of all classes (this study includes two classes, i.e., the picking area and background). *mPA* (%) is calculated using the following equation:

$$mPA = \frac{1}{2} \left( \frac{TP}{TP + FP} + \frac{TN}{TN + FN} \right), \tag{9}$$

*mIoU* (%) is based on the set of pixel points, and represents the ratio of the intersection and union of the predicted and true target regions, evaluating the degree of overlap between the predicted and true target regions. *mIoU* (%) is calculated by using the following equation:

$$mIoU = \frac{1}{2} \left( \frac{TP}{TP + FP + FN} + \frac{TN}{TN + FN + FP} \right), \tag{10}$$

Here, *TP*, *FP*, *TN* and *FN* evaluate each of the pixel points [32].

The volume of the segmentation model is also evaluated using GFLOPS (G) and PARAMS (M). The detection speed is evaluated using *FPS*.

*3.2. Results of Tea Bud Detection Based on YOLOX-Tiny*

3.2.1. Comparison of Different Target Detection Models

In this study, we compared the detection results of YOLOX-tiny with Retinadet, Faster R-CNN, SSD, YOLOv5-s and YOLOX-s. The experiments were performed on the tea bud dataset. The experimental results are shown in Table 4:

**Table 4.** Results of different target detection models.

|  | *mAP* 0.5 (%) | *mR* (%) | *FPS* (Frame) | GFLOPS (G) | PARAMS (M) |
|---|---|---|---|---|---|
| Retinadet -resnet50 | 94.28 | 89.21 | 32.82 | 164.553 | 36.392 |
| Faster R-CNN -vgg | 92.75 | 95.10 | 26.87 | 401.764 | 136.750 |
| SSD-vgg | 86.42 | 77.53 | 92.52 | 274.493 | 24.013 |
| YOLOv5-s | 89.54 | 80.41 | 66.76 | 16.502 | 7.072 |
| YOLOX-s | 97.61 | 95.36 | 60.21 | 26.763 | 8.939 |
| YOLOX-tiny | 97.13 | 94.22 | 66.70 | 15.236 | 5.034 |

As can be seen from the table, Faster R-CNN and Retinadet have relatively high detection accuracy, but the *mAP* and *mR* of both models are lower than YOLOX-s, and their detection speed is slow, where the 26.87 *FPS* of the two-stage network Faster R-CNN does not reach the 30 *FPS* required for real-time detection [33]. Faster R-CNN has the largest computation and parameter amounts. SSD and YOLOv5-s have better detection speed than YOLOX-s and YOLOX-tiny, but they have lower *mAP* and *mR*. YOLOX-s has the highest *mAP* and *mR* values, at least 0.48% and 0.26% higher than other target models, respectively. The detection accuracy of YOLOX-tiny is slightly lower than and almost close to that of YOLOX-s, with *mAP* and *mR* only 0.48% and 1.14% lower than YOLOX-s, respectively. The *mAP* of YOLOX-tiny is only second to YOLOX-s and higher than other models, and *mR* is only lower than YOLOX-s and Faster R-CNN. The computation and parameter amounts are 43.07% and 43.68%, respectively, less than YOLOX-s. YOLOX-tiny is the lightest among the above models and its detection speed is relatively fast. Comprehensively considering all the indicators, YOLOX-tiny has close detection accuracy, lower computational cost and faster detection speed than YOLOX-s, so YOLOX-tiny is more suitable for the needs of this study.

3.2.2. Results of Improvement of YOLOX-Tiny

YOLOX-tiny was selected to detect the tea buds, and in this study, some modifications were made to the YOLOX-tiny to improve its detection performance. The improved YOLOX-tiny is proposed through introducing the *Mish* activate function and a CARAFE module. The results of each step of improvement for YOLOX-tiny are shown in Table 5, where Model (a) is the original YOLOX-tiny model, Model (b) is the YOLOX-tiny with the *Mish* function as the activation function, and Model (c) is the final improved model, with the *Mish* function as the activation function and a CARAFE module to implement the upsampling operation. The experiments were also performed on the tea bud dataset. The detection results of each model were compared to analyze whether the incorporation of each module achieves positive results and to verify the feasibility of the improved method in this study.

**Table 5.** Results of YOLOX-tiny with different modules added.

|  | *mAP* 0.5 (%) | *mR* (%) | *FPS* (Frame) | GFLOPS (G) | PARAMS (M) |
|---|---|---|---|---|---|
| (a) | 97.13 | 94.22 | 66.70 | 15.236 | 5.034 |
| (b) | 97.36 | 94.49 | 63.30 | 15.236 | 5.034 |
| (c) | 97.42 | 95.09 | 59.82 | 15.730 | 5.214 |

As can be seen from the table, the results of each step of improvement are positive, with the addition of each module improving the detection results of the model. Compared to the original model (Model (a)), the *mAP* and *mR* values of the final improved model (Model (c)) are improved by 0.29% and 0.87%, respectively, with a slight increase in computation and parameter amounts, which increased by 0.494 G and 0.18 M, respectively. The detection speed decreased slightly, to 59.82 *FPS*. The *mAP* and *mR* values of the final improved YOLOX-tiny are 97.42% and 95.09%, respectively, and the computation and parameter amounts are 15.730 G and 5.214 M, respectively, which meets the requirements for precise tea bud detection. In addition, the detection speed of 59.82 *FPS* meets the requirements for real-time detection. The loss curve during training of the improved YOLOX-tiny (Model (c)) is shown in Figure 7, the model reaches convergence after approximately the 250th epoch.

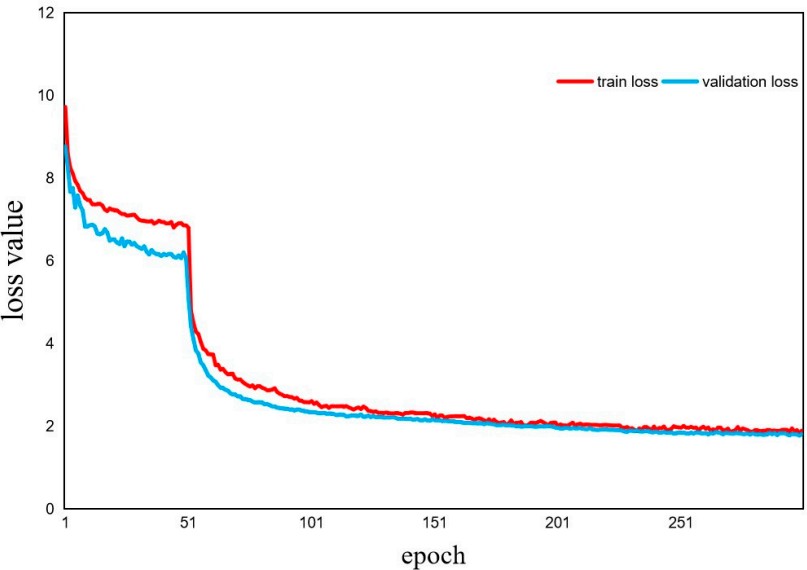

**Figure 7.** The loss curve of the improved YOLOX-tiny during training.

### 3.3. Results of Picking Area Segmentation Based on PSP-Net

3.3.1. Comparative Study of Different Semantic Segmentation Models

The picking areas within the tea bud detection boxes were segmented through the semantic segmentation model. To select the most suitable model, performed on the picking area dataset, the segmentation results of semantic segmentation models such as PSP-net, U-net, DeeplabV3+ and Hrnet were compared, and the results are shown in Table 6.

**Table 6.** Results of different semantic segmentation models.

|  | *mIoU* (%) | *mPA* (%) | *FPS* (Frame) | **GFLOPS (G)** | **PARAMS (M)** |
|---|---|---|---|---|---|
| U-net | 90.48 | 91.41 | 36.31 | 184.167 | 43.933 |
| Hrnet-18 | 79.00 | 81.95 | 14.68 | 37.337 | 9.637 |
| DeeplabV3+ | 85.45 | 89.84 | 18.73 | 166.858 | 54.709 |
| PSP-net | 87.98 | 89.95 | 28.92 | 118.429 | 46.708 |

U-net achieves the highest accuracy for both *mIoU* and *mPA*, but it has the highest computational complexity and requires more computing power from the device. The *mIoU*, *mPA* and *FPS* values of PSP-net are 87.98%, 89.95% and 28.92, respectively, compared to Hrnet and DeeplabV3+; the *mIoU* of PSP-net is 8.98% and 2.53% higher and *mPA* is 8.00% and 0.11% higher, respectively. In terms of volume, Hrnet is the lightest model, but Hrnet also achieves the lowest segmentation accuracy and detection speed. PSP-net has lower computation and parameter amounts than DeeplabV3+, while its detection accuracy and

speed are both better than DeeplabV3+. In summary, considering both accuracy and speed, PSP-net is the most suitable model for this study.

### 3.3.2. Results of PSP-Net with Different Modules Added

The volume of the original PSP-net model was too large; therefore, in this study, some modifications were made to the PSP-net to make it more relevant to the needs of the segmentation task. The PSP-net model was modified by replacing its backbone network with MobileNetV2 and replacing the $3 \times 3$ convolution in its pyramid pooling module with ODConv, and the results for each step of modification are shown in Table 7, where Model (a) is the original PSP-net, Model (b) is the first step modification to the PSP-net, namely the PSP-net with its backbone network replaced with MobileNetV2, and Model (c) is the final improved PSP-net, namely the PSP-net with both the backbone network replaced with MobileNetV2 and the $3 \times 3$ convolution replaced with ODConv. The experiments were performed on the picking area dataset.

**Table 7.** Results of each step of modification for PSP-net.

|  | *mIoU* (%) | *mPA* (%) | *FPS* (Frame) | **GFLOPS (G)** | **PARAMS (M)** |
|---|---|---|---|---|---|
| (a) | 87.98 | 89.95 | 28.92 | 118.429 | 46.708 |
| (b) | 87.81 | 91.97 | 88.03 | 6.031 | 2.376 |
| (c) | 88.83 | 92.96 | 84.90 | 5.086 | 1.821 |

After replacing the backbone network with MobileNetV2, the *mIoU* is reduced by 0.17%, the *mPA* is improved by 2.02%, the computation and parameter amounts are both reduced by 94.91%, and the detection speed is significantly improved to 88.03 FPS. After replacing the $3 \times 3$ convolution with ODConv on the basis of the previous modification, the *mIoU* and *mPA* are improved by 1.02% and 0.99%, respectively, and the computation and parameter amounts are further reduced by 0.945 G and 0.555 M, respectively, with a slight decrease in detection speed to 84.90 *FPS*. The final improved PSP-net improves the *mIoU* and *mPA* by 0.85% and 3.01%, respectively, compared with the original model, and the computation and parameter amounts are decreased by 95.71% and 96.10%, respectively. The loss curve during training of the improved PSP-net (Model (c)) is shown in Figure 8; the model reaches convergence after approximately the 150th epoch.

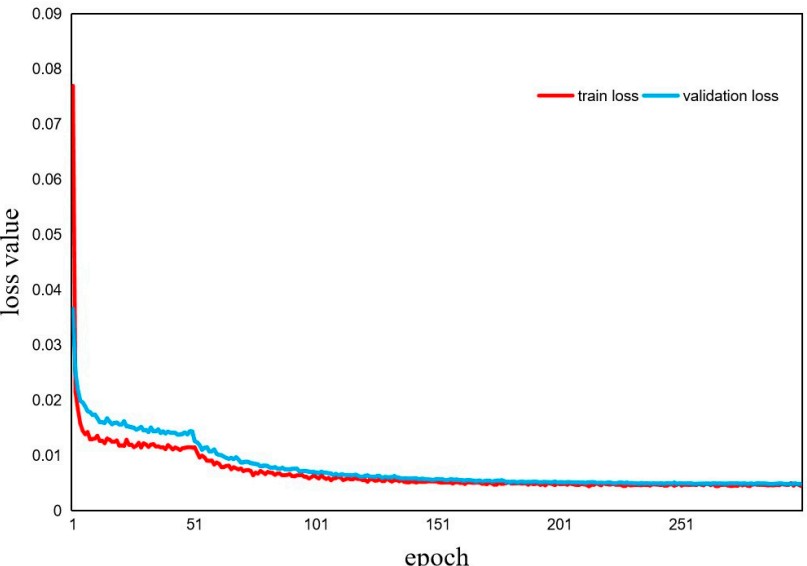

**Figure 8.** The loss curve of the improved PSP-net during training.

### 3.4. Comparison of Our Method with Directly Segmenting the Picking Area

In order to verify that the method proposed in this study has a better detection capability compared with the method of directly segmenting the picking area on the original images, we compared the results of our method with those of directly segmenting the picking area through segmentation algorithms. The improved PSP-net and Mask R-CNN were used for comparison, and to verify that it was necessary to detect tea buds first, in this study, we tried to segment the picking areas directly on the original images through the semantic segmentation model. To make the result equitable and comparable, the improved PSP-net was selected for comparison. However, the approach of finding the interest region of the target first and then performing semantic segmentation, was similar to the idea of the instance segmentation algorithm. The difference is that the instance segmentation algorithm takes the surrounding area of the picking area as the region of interest [34], and our method takes the tea bud area as the region of interest. To verify that our method achieves better performance than the instance algorithm, our method was also compared with the result of directly segmenting the picking area through the instance segmentation model Mask R-CNN at the same time. In this chapter, the experiments of the improved PSP-net and Mask R-CNN were both trained on the directly detecting picking area dataset. To compare the picking area segmentation performance, on the original images, the segmented picking areas obtained by using the three methods, the annotated truth picking areas were compared and their *mIoU* and *mPA* valus were calculated. The indicators *mIoU* and *mPA* merely represented the quality of picking area segmentation. Whereas, in the cases where the result was poorly segmented but the segmented area was still within the truth picking area, the centroid of the segmented area could still be an available picking point, thus *mIoU* and *mPA* could not represent the quality of picking point detection. To compare the picking point detection performance of the three methods, in this study, the indicator of detection rate was introduced. A total of 50 images were randomly selected from the original images, and the picking areas of the 50 images were segmented using our method, the improved PSP-net and Mask R-CNN models. As shown in Figure 9, when the picking area was correctly segmented and the centroid of the segmented area was at the suitable position for picking, the picking point was marked as successfully detected. The ratio of successfully detected picking points to all target picking points is the detection rate. The detection rate of each method was calculated from all selected 50 images. The comparison results are shown in Table 8.

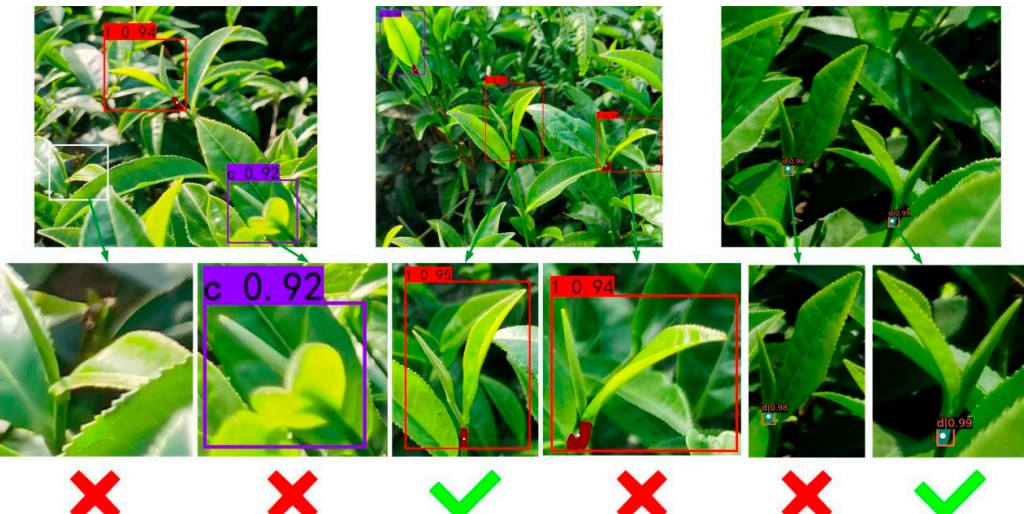

**Figure 9.** Example for judging the successfully detected and wrongly detected picking points. When the detected picking point is marked at the right position suitable for picking, it is marked as successfully detected. When no tea buds or picking areas are detected, or when the picking point is marked at an inaccurate position, it is marked as falsely detected.

**Table 8.** Comparisons of results of our method and the methods of directly detecting the picking area.

|  | *mIoU* (%) | *mPA* (%) | *FPS* (Frame) | Detection Rate (%) |
|---|---|---|---|---|
| Improved PSP-net | 60.54 | 61.30 | 81.77 | 17.2% |
| Mask R-CNN | 58.30 | 59.25 | 10.21 | 21.7% |
| Our method | 83.27 | 86.51 | 32.86 | 95.6% |

The segmentation accuracy of our method is better than that of the other two models, and the *mIoU* and *mPA* of our method are 22.73% and 25.21% higher than those of the improved PSP-net, respectively, and 24.97% and 27.26% higher than those of Mask R-CNN, respectively. In terms of detection speed, the speed of our method after combining two models is 32.86 *FPS*, which meets the requirement of 30 *FPS* for real-time detection. The segmentation accuracy and speed are both higher than those of Mask R-CNN. Although the detection speed of our method is lower than that of the improved PSP-net model, the segmentation accuracy of our method is significantly higher. In terms of the picking point detection rate, Mask R-CNN has a higher detection rate than the improved PSP-net, and the detection rate of our method reaches 95.6%, which is significantly higher than that of the methods directly detecting the picking area.

*3.5. Actual Detection Effects*

3.5.1. Comparison of Actual Detection Effects of the Improved YOLOX-Tiny with the Original YOLOX-Tiny

The actual detection effects of YOLOX-tiny and its improved model are shown in Figure 10. The images under two lighting conditions, i.e., strong light and weak light, are compared. For the images of Group (a) and Group (b) under strong light, the original model misses one Class "1" target, respectively, in both groups, as both targets are partially obscured and the shape of the tea bud is not fully displayed, resulting in the original model failing to identify them. However, the improved model accurately recognizes the two Class "1" targets on the premise of accurately recognizing other targets in the images. For the images of Group (c) and Group (d) under weak light, there are one and two FP results, respectively, falsely detected by the original model, that is, old leaves are misjudged as tea buds. All three FP targets are at the edge of the image, with only part of leaf or tea stem inside the image, showing the trait of one bud with one leaf, so they are all identified as Class "1" targets. The improved model accurately identifies all TP results in the images and correctly eliminates these three FP results. According to the actual detection effects, it can be seen that the improved YOLOX-tiny model outperforms the original model under both strong and weak light conditions, with no significant influence caused by lighting conditions on its detection effects. The improved YOLOX-tiny model is not only able to accurately identify tea buds that are partially obscured, but also the backgrounds that are incorrectly identified as tea buds by the original model, and the improved model is able to accurately identify and reject similar false detection results. In summary, the improved model can accurately identify and correctly classify the tea buds in the images, with low missed and false detection rate, high recognition rate and *recall* rate. The excellent detection capability of the improved YOLOX-tiny can meet the requirements for tea bud target detection in this study.

3.5.2. Comparison of Segmentation Effects of the Original PSP-Net and the Improved PSP-Net

Comparing the segmentation effects of the original PSP-net and the improved PSP-net, the tea bud images obtained from the improved YOLOX-tiny were segmented and the results are shown in Figure 11.

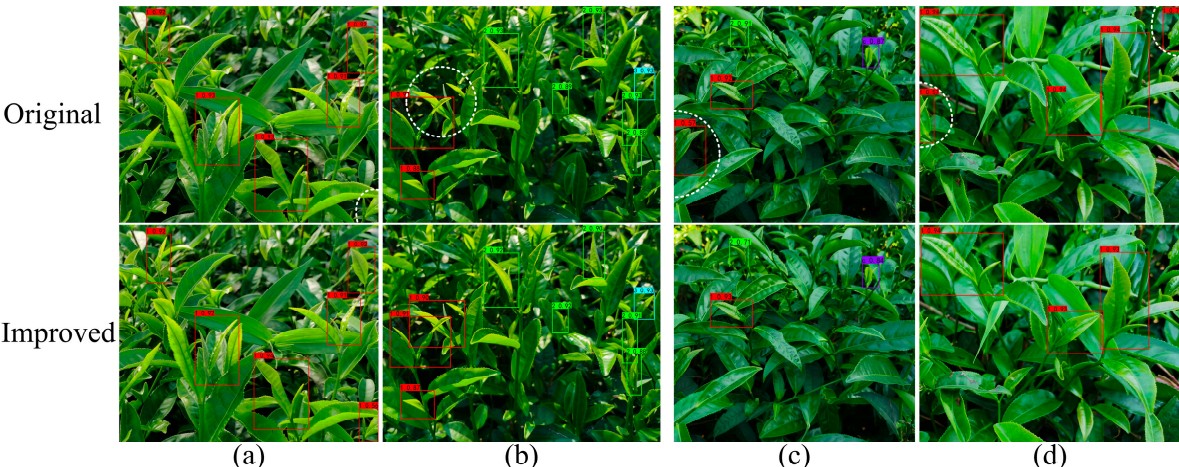

|  |  |  |  |  |
| --- | --- | --- | --- | --- |
| Original |  |  |  |  |
| Improved |  |  |  |  |
|  | (a) | (b) | (c) | (d) |

**Figure 10.** Comparison of actual detection effects of YOLOX-tiny and the improved model: (**a**,**b**) Images under strong light; (**c**,**d**) images under weak light.

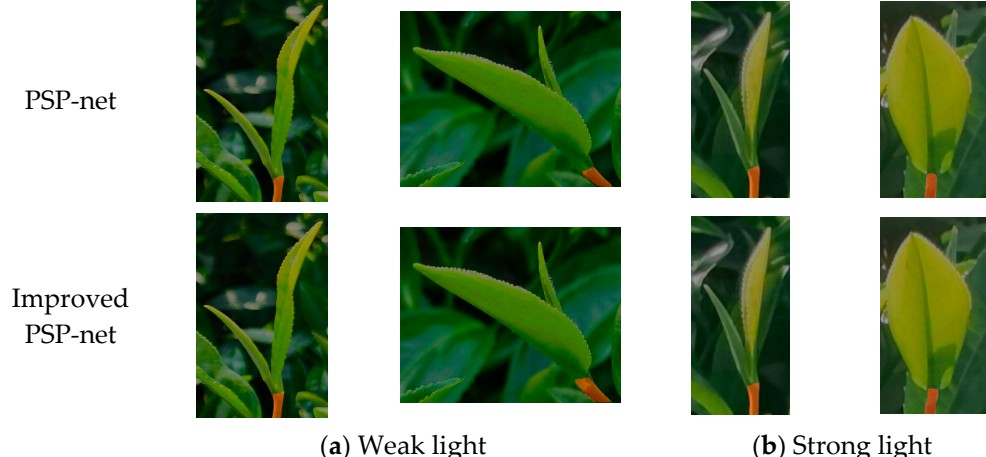

(**a**) Weak light     (**b**) Strong light

**Figure 11.** Comparison of the segmentation effects of PSP-net and the improved model; (**a**) Images under weak light; (**b**) images under strong light.

The original PSP-net and the improved PSP-net both have excellent segmentation effects for the tea stem area within the detection box. With a significant reduction in the computation and parameter amounts (computation amount decreases by 95.71% and parameter amount decreases by 96.10%), the improved PSP-net has close segmentation effects to the original model and is able to accurately identify the tea stem area and segment the pixel points of the picking area, with few misjudged background pixels. In addition, comparing the images under strong and weak light shows that the performance of the improved model is not affected by the lighting conditions. The segmentation capability of the improved PSP-net model can meet the requirements for identifying the picking areas in this study.

### 3.5.3. Comparison of Actual Picking Area Segmentation Effects of Our Method and the Methods of Directly Segmenting the Picking Area

We compare the actual picking area segmentation effects on the original images using our method and using the semantic segmentation model (the improved PSP-net) and the instance segmentation model (Mask R-CNN); the results are shown in Figure 12.

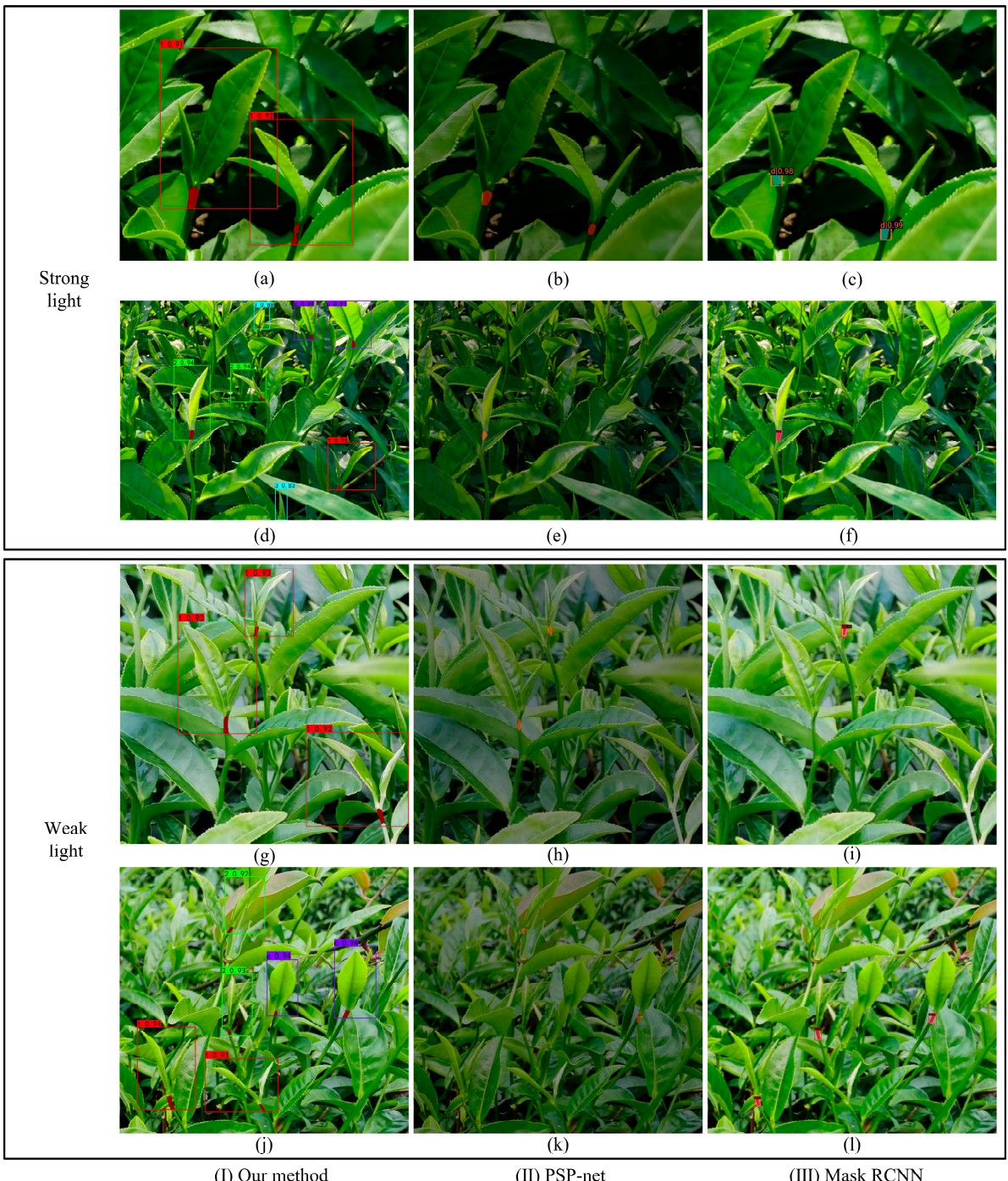

Strong light

Weak light

(I) Our method　　　　(II) PSP-net　　　　(III) Mask RCNN

**Figure 12.** Actual picking area segmentation effects on the original images of our method compared with the methods directly segmenting the picking area. Subfigures (**a**–**f**) are images under strong light, subfigures (**g**–**l**) are images under weak light. Subfigures (**a**,**d**,**g**,**j**) are results of our method, subfigures (**b**,**e**,**h**,**k**) are results of PSP-net, subfigures (**c**,**f**,**i**,**l**) are results of Mask RCNN.

The performance of our method is significantly better than the other two algorithms. For images with fewer targets under strong light (Figure 12a–c), all three methods can correctly identify two targets. The segmentation result of the improved PSP-net is rough and contains part of background pixels, the result of Mask R-CNN is more accurate, whereas one of the picking areas (shown in Figure 12c) is wrong and the segmented picking area is for a single bud, not for the one bud with one leaf required in this study. Our method accurately identifies the two Class "1" targets and classifies them correctly, but one of the

segmented picking areas is incomplete. For images with multiple targets under strong light (Figure 12d–f), the improved PSP-net and Mask R-CNN methods both show a large number of missed targets, and the improved PSP-net only identifies two targets, which still has the problem of rough segmentation results. The segmentation result of Mask R-CNN is more accurate, but only one target is identified by Mask R-CNN. Our method accurately identifies, classifies and segments all the targets. However, an obscured Class "3" target is detected, but the tea stem part is not in the image, so it is not segmented. For images with fewer targets under weak light (Figure 12g–i), the improved PSP-net and Mask R-CNN, respectively, lose one and two targets, and our method successfully detects all the targets. For images with multiple targets under weak light (Figure 12j–l), the improved PSP-net and Mask R-CNN methods lose a large number of targets, whereas our method has no target missing phenomenon, accurately identifies and correctly classifies all targets in the image, and the segmentation effects are also ideal; therefore, it is able to completely and accurately segment the picking area. The comparison shows that the segmentation ability for the picking areas on the original images of the improved PSP-net and Mask R-CNN are greatly affected by the lighting conditions and the number of targets. In the case of weak light or multiple targets, the two models tend to miss a large number of targets, resulting in poor picking area segmentation effects and low detection rate. The method proposed in this study has significantly better performance under strong light and weak light with multiple targets or fewer targets.

### 3.5.4. Picking Point Marking

After using the improved YOLOX-tiny to detect the tea bud boxes and using the improved PSP-net to segment the picking area within the tea bud detection box, the centroid of the picking area is taken as the picking point. As shown in Figure 13, the centroid of the picking area falls exactly on the tea stem and is at a moderate distance from the end of tea buds, which is suitable for picking. This method for picking point selection is appropriate and meets the requirements for actual picking.

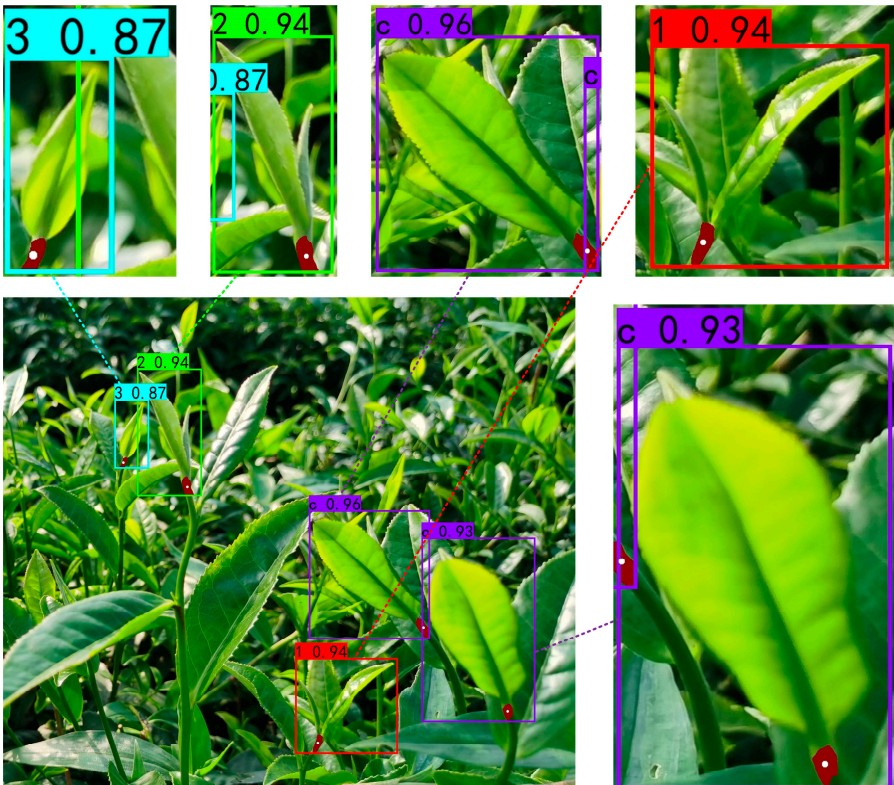

**Figure 13.** The actual picking point selection effects.

## 4. Discussion

Famous tea is still dependent on manual picking, and intelligent picking should be applied to the industry as soon as possible. In the actual situation, the color of tea buds in complex scene is close to that of the background, and the target is small, making it difficult to distinguish. Picking point selection is built on the basis of tea bud identification, with even smaller targets and no clear endpoints to choose from, causing greater difficulties for intelligent picking. To address this problem, in this study, we creatively proposed a method that combined target detection and semantic segmentation networks, to detect the location of tea buds first, and then performed accurate picking point extraction. As can be seen from the above figures, unlike most other studies, the Fuyun 6 tea species dataset used in this study contains all growth stages from tender to mature, and the tea buds are in different postures, such as lateral positions, which is more in line with the actual working conditions of picking. The experimental results show that our method performs better than the methods directly detecting picking areas on the original images, with a low missing rate, high detection rate and accurate picking point location. The *mIoU* and *mPA* of this method for picking area segmentation are 83.27% and 86.51%, respectively, and the detection rate of picking points reaches 95.6%.

In this study, the lightweight YOLOX-tiny was chosen to detect tea buds. The computation and parameter amounts of YOLOX-tiny are, respectively, 56.93% and 56.32% those of YOLOX, but its detection accuracy is close to that of YOLOX. There are only four classes in the tea bud detection dataset, and the number of images is not large, with a total of 2126. The YOLOX-tiny model with smaller depth and width is more suitable for this dataset to avoid wasting computation power caused by excessive computational complexity. In order to facilitate the deployment of our method to an outdoor mobile device during actual picking, the lightest possible model should be chosen without significant loss of accuracy. Therefore, the YOLOX-tiny with more lightweight volume and faster detection speed is more suitable for the needs of this study. To improve the detection accuracy of the model, the activation function of the original model was replaced by the *Mish* function to obtain better nonlinear expression capability, and its upsampling operation was implemented using a CARAFE module to increase the perceptual domain and to enhance the usage of contextual semantic information, avoiding losing more feature information and more deeply understanding the content of the features. The two modifications for YOLOX-tiny effectively improve the detection capability of the model.

The picking area was obtained by segmenting the tea stem area within the cropped tea bud detection boxes. Each box contains only one tea stem area; there was a single class of target, so the segmentation task was relatively simple. The original PSP-net was too computationally intensive, and the detection speed of 28.92 *FPS* did not meet the 30 *FPS* required for real-time detection. Replacing the PSP-net backbone network with the lightweight MobileNetV2 resulted in a significant reduction in the computation and parameter amounts and a significant increase in detection speed, yet only caused a slight change in accuracy, with a 0.17% reduction in *mIoU* and a 2.02% increase in *mPA*, indicating that MobileNetV2 with higher computational efficiency is more suitable for a simple computational environment. After the ODConv was added, the *mIoU* reached 88.83%, surpassing that of the original PSP-net, and the *mPA* was further improved to 92.96%. The ODConv learns complementary attention along four dimensions of the kernel space using a multidimensional attention mechanism through a parallel approach. The results show that the linear combination of its convolution kernels and associated attention can enhance the feature expression of the model and can improve the accuracy of the lightweight convolutional network. The final improved PSP-net decreases the computation and parameter amounts by 95.71% and 96.10%, respectively, compared to the original model; improves the *mIoU* and *mR* by 0.85% and 3.01%, respectively; and significantly increases the detection speed to 84.90 *FPS*. The improved PSP-net significantly improves the lightweight characteristics of the model while improving the segmentation capability in

all aspects. The results show that the improved PSP-net has good segmentation capability for the picking area within the detection box.

The areas suitable for picking are small and densely distributed, and the color is very similar to the background. When directly segmenting the picking area without preprocessing of the improved YOLOX-tiny, the segmentation models (the improved PSP-net and Mask R-CNN) cannot extract enough features, resulting in low segmentation accuracy and poor effects. Without reference to the tea bud detection box, the picking areas of a single bud and one bud with one leaf are relatively similar in appearance, and Mask R-CNN is prone to misidentify the picking location, as shown in Figure 12c, where the picking area of a single bud is misidentified as that of one bud with one leaf, the similar mistakes are prone to cause wrong picking, affecting the quality of the product. The improved PSP-net can more accurately identify the picking position of one bud with one leaf. The *mIoU* and *mPA* of Mask R-CNN are lower than those of the improved PSP-net, but its detection rate of picking points is higher than that of the improved PSP-net, which is because, if the model successfully segments part of the pixel areas within the actual picking area, then the centroid must be located at a suitable position for picking and the case is judged as a successful detected picking point. Therefore, the result of detection rate depends more on the number of picking areas segmented by the model, and less on the segmentation quality of a single area. Mask R-CNN identifies more picking areas, so its picking point detection rate is higher. The detection effects of both models are largely influenced by the lighting conditions and the number of targets; in scenarios with weak light and multiple targets, the improved PSP-net and Mask R-CNN miss a large number of targets, and the two models fail to segment the picking area directly on the original images. Our method first identifies tea buds through the target detection model. Tea bud targets generally occupy a larger area on the original image than picking areas and look more conspicuous, making it less difficult to identify tea buds. Detecting tea buds first can significantly reduce the target missing rate, and therefore, the detection rate is higher than that of directly identifying the picking area. Segmenting the picking area within the tea bud detection box, limits the picking position to one bud with one leaf, which can avoid the phenomenon of wrong position of segmented area, and significantly reduces the background complexity, and thus, difficulty of the segmentation task. Thus, the *mIoU* and *mR* of the improved PSP-net on picking area segmentation within the detection boxes are much higher than those of the same model on picking area segmentation on the original images. The results show that our method has much better detection capability than that of the methods directly segmenting the picking area; the accuracy of tea bud detection and picking area segmentation are both relatively high, which can accurately detect and correctly classify tea bud targets, and then segment picking areas more accurately. Our method has excellent detection results for images with few or multiple targets under both strong and weak light. The disadvantage of combining the two models of the improved YOLOX-tiny and the improved PSP-net is a significant loss in detection speed, but 32.86 *FPS* still meets the requirement for real-time detection.

During actual picking, a robotic arm requires precise coordinates, so the pixel region of the picking area is not enough, and the picking point coordinates are needed. In this study, the picking point is extracted on the basis of the segmented picking area. The picking point needs to be located at a suitable position on the tea stem; the distance to the bud must not be too close, which could damage the bud during picking, nor too far, which could affect the economic value of the finished product if the buds are with a too long stem. The centroid of the picking area is chosen as the picking point, this method is suitable for tea bud picking, efficiently controlling the distance from the picking point to the two ends of the picking area, and the distance to the tea bud is kept moderate. The segmentation result of the picking area is not always extremely accurate, whereas this picking point selection method has a certain fault-tolerant space for that, as shown in Figure 12a; if the segmented area is not complete, the picking area does not contain all the tea stem pixel points, which does not seriously affect the position of its centroid point, if the segmented area is too large, the picking area contains a small portion of background pixel points, the centroid point still



falls on the tea stem. When the stem part is obscured, as shown in Figure 12d, the improved YOLOX-tiny recognizes the Class "3" target, but its stem part is not within the image; thus, the improved PSP-net does not segment it, and the wrong position information would not be returned, causing no adverse effects, this is somehow a screening mechanism.

The research object of this study is only the Fuyun 6 tea species. Future research could add datasets of other tea species to enhance the detection ability of our method on other varieties to improve its applicability and robustness. In addition, this study only researched the one bud with one leaf grade with high economic value. For the needs of different markets, the single bud and one bud with two leaves grades are also worthy of studying. Other grades of tea buds could also be added in order to enhance the detection and classification ability of this method for different tea species and grades of tea buds, so as to develop a multifunctional intelligent tea picking robot in one collection.

## 5. Conclusions

To address the difficulty of detecting tea buds and the picking points in complex backgrounds, this study proposed a method combining a target detection model and a semantic segmentation model at the inference end, the improved YOLOX-tiny was used to detect tea buds first, the tea bud detection boxes were input into the improved PSP-net for picking area segmentation within the tea bud detection boxes, and the centroid of the picking area was taken as the picking point. The method achieves an *mIoU* of 83.27% and an *mPA* of 86.51% for picking areas on the original images, and the picking point detection rate is 95.6%, the detection speed satisfies the real-time detection requirements. The actual detection effects show that our method is better than the method directly segmenting the picking area.

This study proposed an improved YOLOX-tiny as the tea bud detector, which was modified by replacing the activation function with the *Mish* function, and introducing a CARAFE module to implement the upsampling operation. The *mAP* of the improved YOLOX-tiny is 97.42% and the *mR* is 95.09%, which are improved by 0.29% and 0.87%, respectively, compared with the original model. The actual detection effects show that the improved model performs better than the original model under both strong and weak light.

This study proposed an improved PSP-net for picking area segmentation within the detection box, which was modified by replacing the backbone network with MobileNetV2 and the conventional convolution with ODConv. The *mIoU* and *mPA* of the improved model are 88.83% and 92.96%, which are improved by 0.85% and 3.01%, respectively, and the computation and parameter amounts decrease by 95.71% and 96.10%, respectively, compared to those of the original PSP-net.

Datasets of other tea species and other grades of tea buds such as the single bud and the one bud with two leaves can be added in future research to enhance the detection and classification ability of this method for different tea species and grades of tea buds.

**Author Contributions:** Conceptualization, J.M. and F.K.; methodology, J.M.; software, J.M. and J.Z.; validation, S.T., C.C. and C.Z.; formal analysis, J.M.; investigation, J.M. and Y.A.; resources, J.M.; data curation, Y.A.; writing—original draft preparation, J.M. and Y.A.; writing—review and editing, J.M. and F.K.; visualization, Y.W.; supervision, Y.W.; project administration, Y.W. and F.K.; funding acquisition, Y.W. All authors have read and agreed to the published version of the manuscript.

**Funding:** This research was funded by the Ningxia Hui Autonomous Region key research and development plan project, grant number 2022BBF01002.

**Data Availability Statement:** All data generated or presented in this study are available upon request from the corresponding author. Furthermore, the models and code used during the study cannot be shared at this time as the data also form part of an ongoing study.

**Conflicts of Interest:** The authors declare no conflict of interest.

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
