# Peer review of "Tea Bud and Picking Point Detection Based on Deep Learning"

_forests, doi:10.3390/f14061188_

Round 1

Reviewer 1 Report

Junquan et al. developed a method for tea bud and picking point detection through the use of YOLOX-tiny model and PSP-net model. Small modification of these two models were performed to make the method lightweighted and accurate. From the results presented, the goal was achieved well. Meanwhile, the manuscript was carefully prepared. Thus, I agree with its acceptance of current version.

Reviewer 2 Report

In this paper, the authors present "Tea bud and picking point detection based on deep learning." The PSP-net was modified by replacing its backbone network with the lightweight network MobileNetV2, and replacing the conventional convolution in its feature fusion part with Omni-Dimensional Dynamic Convolution. The model was greatly lightweighted and its segmentation accuracy for the picking area was improved. The mean Intersection over Union and mean Pixel Accuracy of the improved PSP-net model are 88.83% and 92.96%, respectively, while its computation and parameters amounts are reduced by 95.71% and 96.10% compared to the original PSP-net. The method proposed in this study achieves a mean Intersection over Union and mean Pixel Accuracy of 83.27% and 86.51% for the overall picking area segmentation, and the detecting rate of picking point identification reaches 95.6%. Moreover, its detection speed satisfies the requirements of real-time detection, providing a theoretical basis for the automated picking of famous tea. However, there are some issues should be addressed.

1. The approach of finding the interest region of the target first and then performing semantic segmentation, was similar to the idea of the instance segmentation algorithm.  What is the innovation of the proposed method?

2. An obscured Class “3” target is detected, but the tea stem part is not in the image, so it is not segmented. How to solve this problem?

3. The picking of famous tea still relies on artificial methods, with low efficiency, labor shortage and high labor costs, which restricts the development of the tea industry. How to solve this problem?

4. Famous tea has high nutritional and market value and rare production, it strictly requires the quality of picking, so it still relies on the way of manual picking with low efficiency, high cost and labor shortage, which has become important problems restricting the development of famous tea industry. How to overcome this problem in this paper?

5. Tea buds in natural state is complex and the difficulties to identify the bud and select the picking point. How to solve this problem?

6.For images with multiple targets under strong light (Figure 9 (d)-(f)), the improved PSP-net and MaskRCNN both show a large number of missed targets, and the improved PSP-net only identifies two targets, which still has the problem of rough segmentation results. How to overcome this problem in this paper?

Extensive editing of English language required

Reviewer 3 Report

Authors in this study proposed a method based on deep learning, combining object detection and semantic segmentation networks to identify tea bud picking points  Authors  proposed an improved PSP-net semantic segmentation model for segmenting the picking area inside the detection box. The PSP-net was modified by replacing its backbone network with the lightweight network MobileNetV2 and replacing the conventional convolution in its feature fusion part with Omni-Dimensional Dynamic Convolution.

I have the following reservations about the article:

-The abstract should be factual and describe what the authors did in their work and what results they achieved. The abstract should not contain an introduction to the subject.

-The authors in the article talk about 3 datasets divided into training and validation sets. Did the authors also use the test set during the experiment, and if so, in what proportion?

-Figure 2. There are two different images with different dimensions. It would be good if the authors modified this figure to consist of two equally sized images.

-Chapter 3.1 is a chapter with theoretical knowledge, but it does not contain any references, which could be considered plagiarism.

-The authors detail several evaluation metrics but use only a few. If the authors use only a few of all the metrics described, it would be good to explain why they have chosen the metrics mentioned.

-The results are very vaguely described and chaotically treated in the paper. For example. Tables 4 and 5 show the results of different networks, but in Table 5 authors discuss the results of differently modified YOLOX-tiny networks, while in Table 4 authors discuss the results of YOLOX-tiny. It is not clear in the paper which network authors mean.

-Also, the results are in Table 6, and it is not clear which dataset the authors use. At the beginning, they mention that they work with three datasets, but then it is not explained which dataset the outliers are on.

-In Tables 6 and 7, it happens again that in both tables there are PSPnet results, but each one is compared with other networks. 

-In Table 8, the authors get to their results, but the information on which dataset and on which data the results are obtained and whether they are comparable with other networks and not only with the ones in the table is lost. In this table, the authors have suddenly changed the metrics, it is also not clear why. 

-The results are very unclear and need to be redone.

-The authors should add confusion matrices or plots of the training process to the paper.

Round 2

Reviewer 2 Report

The authors have solved the related problems. It is good enough.

Extensive editing of English language required.

Reviewer 3 Report

The authors have answered all questions and incorporated all comments